# Hemin Ameliorates the Inflammatory Activity in the Inflammatory Bowel Disease: A Non-Clinical Study in Rodents

**DOI:** 10.3390/biomedicines10082025

**Published:** 2022-08-19

**Authors:** Inês Silva, Rita Correia, Rui Pinto, Vanessa Mateus

**Affiliations:** 1H&TRC–Health and Technology Research Center, ESTeSL–Escola Superior de Tecnologia da Saúde de Lisboa, Instituto Politécnico de Lisboa, Av. D. João II, Lote 4.69.01, 1990-096 Lisbon, Portugal; ines.silva@estesl.ipl.pt (I.S.); ritaaralopescorreia@gmail.com (R.C.); 2iMed.ULisboa, Faculdade de Farmácia, Universidade de Lisboa, 1990-096 Lisboa, Portugal; rapinto@ff.ulisboa.pt; 3Joaquim Chaves Saúde, Dr. Joaquim Chaves, Laboratório de Análises Clínicas, Miraflores, 1495-069 Algés, Portugal

**Keywords:** hemin, inflammatory bowel disease, experimental colitis, TNBS-induced colitis, anti-inflammatory effect, heme-oxygenase inducer

## Abstract

Background: Inflammatory bowel disease (IBD) is a chronic inflammatory disorder of the gastrointestinal tract. Currently, there is no cure, and pharmacological treatment aims to induce and maintain remission in patients, so it is essential to investigate new possible treatments. Hemin is a heme-oxygenase inducer which can confer anti-inflammatory, cytoprotective, and antiapoptotic effects; therefore, it can be considered an asset for different gastrointestinal pathologies, namely for IBD. Aim: This experiment aims to evaluate the efficacy and safety of hemin, in a chronic 2,4,6-trinitrobenzenesulfonic acid (TNBS)-induced colitis model in rodents. Methods: The induction of chronic colitis consisted of five weekly intrarectal administrations of 1% TNBS. Then, the mice were treated daily with 5 mg/kg/day or 10 mg/kg/day of hemin, through intraperitoneal injections, for 14 days. Results: Hemin demonstrated an anti-inflammatory effect through the reduction in tumor necrosis factor (TNF)-α levels, fecal calprotectin, and fecal hemoglobin. It was also found to be safe in terms of extraintestinal manifestations, since hemin did not promote renal and/or hepatic changes. Conclusions: Hemin could become an interesting tool for new possible pharmacological approaches in the management of IBD.

## 1. Introduction

IBD is characterized as a chronic and intermittent inflammatory response localized in the gastrointestinal tract, which can be represented by two main phenotypes, Crohn’s disease and ulcerative colitis [1,2,3,4,5]. Both conditions are autoimmune disorders that are not medically curable [6]. This chronic disease has become a world health problem in the 21st century, with accelerating incidence in newly industrialized countries [6,7]. Currently, it is estimated that nearly 3.9 million females and nearly 3.0 million males are living with IBD worldwide, and there is an upward trend of new cases [8]. Some gastrointestinal signs and symptoms are diarrhea, abdominal pain, and hematochezia. Additionally, it can also cause extraintestinal manifestations to develop, such as anemia, fever, weight loss, arthritis, sclerosing cholangitis, uveitis, pyoderma gangrenosum, and erythema nodosum [9,10,11]. Currently, the therapeutic strategies for patients with IBD are related to modifications in lifestyle habits, and pharmacological and surgical treatments. Indeed, the therapy consists of taking aminosalicylates, glucocorticoids, immunosuppressants, immunomodulators, and biologic therapy. These drug treatments aim to induce and maintain the patient in remission and ameliorate the disease´s secondary effects, rather than modifying or reversing the underlying pathogenic mechanism [11]. Therefore, it is essential to investigate new potential pharmacological approaches.

Hemin, or ferriprotoporphyrin IX chloride, is a metalloporphyrin containing iron, currently used for acute cases of porphyria [12,13]. Hemin is known to induce the enzyme heme-oxygenase (HO) [14]. HO is a rate-limiting enzyme for heme catabolism, a process that leads to the production of biliverdin, free iron, and carbon monoxide [15]. Three HO isoenzymes have been identified in mammals, namely HO-1, HO-2, and HO-3 [16]. HO-1 is an enzyme that is involved in the catabolism of the heme group, having a cytoprotective action capable of inhibiting the inflammatory response and reducing oxidative stress [17].

HO-1 expression can be induced by hemin. Proteins that contain a heme group play a fundamental role in the physiological processes related to oxygen transport, mitochondrial respiration, and signal transduction. Since the free heme group exerts a cytotoxic action by forming free oxygen radicals and promoting lipid peroxidation, this group is degraded by the enzyme heme-oxygenase. This enzyme is expressed in the constitutive form, HO-2, which regulates the normal functioning of cells, and in the induced form, HO-1, which is expressed as a response to tissue damage. Degradation of the heme group by HO-1 will result in equimolar amounts of carbon monoxide (CO), Fe^2+^, and biliverdin. Later, the biliverdin is reduced to bilirubin by biliverdin reductase [18]. It has been shown that HO-1 and HO-1 mRNA expression is remarkably elevated in situations of intestinal inflammation when compared to healthy intestines. Furthermore, in situations of mucosal inflammation in active ulcerative colitis, HO-1 protein expression is also elevated. This evidence suggests that the increase in HO-1 expression is due to inflammation. HO-1 expression occurs essentially in macrophages, as in previous histological studies the result was CD-68 positive [18]. Although the expression is mostly observed in macrophages, other studies have described the localization of HO-1 in the colonic mucosa, which, in addition to being expressed in inflammatory cells, is also expressed in epithelial cells.

Currently, there is already preclinical evidence demonstrating the anti-inflammatory properties of hemin in the context of IBD, through the significant attenuation of the inflammatory response and extraintestinal manifestations [16]. However, that experiment was carried out in 4 days, leading us to be interested in evaluating the effect of hemin on colitis from a chronic perspective, considering its long-term application. Therefore, the main objective of this experiment is to evaluate the efficacy and safety of the administration of hemin, in a model of chronic colitis induced by TNBS.

## 2. Materials and Methods

### 2.1. Chemicals

TNBS 5%, ferriprotoporphyrin IX chloride (hemin), and sodium hydroxide (NaOH) were acquired from Sigma Chemical (Sintra, Portugal). Ketamine (Imalgene^®^ 1000) and xylazine (Rompun^®^ 2%) were obtained from Merial (Lisbon, Portugal) and Bayer (Lisbon, Portugal), respectively. An ADVIA^®^ kit was purchased from Siemens Healthcare Diagnostics (Erlangen, Germany). Enzyme-linked immunosorbent assay (ELISA) kits for fecal calprotectin measurements were acquired from Hycult Biotechnology (Uden, Netherlands).

### 2.2. Animals

Female CD-1 mice (6 weeks old, 30 g ± 7 g body weight) were used for the experiment. Animals had one week for acclimatization and were housed in standard polypropylene cages, with access to food and water ad libitum in the bioterium of the Faculty of Pharmacy, University of Lisbon. The temperature, humidity, and lighting were controlled throughout the study. The mice were kept at 18–23 °C and 40–60% humidity, in a controlled 12 h day/night cycle. Animal care was in accordance with the internationally accepted principles for laboratory animal use and care, found in Directive 2010/63/EU [19]. The experiment was approved by the Ethics Committee for Animal Experimentation of the Faculty of Pharmacy, University of Lisbon (ORBEA) (code nr. 3/2020) and approved by *Direção Geral de Alimentação e Veterinária* (DGAV) on November 6th of 2020.

### 2.3. Induction of Experimental Colitis

The mice were left unfed for 24 h before induction. First, on day 0 (induction day), the mice were anesthetized with 40 μL of a mixture of ketamine 100 mg/kg + xylazine 10 mg/kg by an intraperitoneal (IP) injection. TNBS was instilled through a single intrarectal (IR) administration [16,20,21,22,23]. The technique consisted of the insertion of a cannula into the colon, such that the tip was 4 cm proximal to the anus, and then 100 μL of TNBS 1.0% was administrated over five weeks. Finally, the mice were kept in a Trendelenburg position for 1 min to avoid reflux. The mice were subjected to five IR administrations of 100 μL of TNBS 1.0%, one per week (days 0, 7, 14, 21, and 28 of the experiment).

On the last day of the experiment, day 35, the animals received another IP administration of 40 μL of ketamine 100 mg/kg + xylazine 10 mg/kg mixed in order to be anesthetized. Then, a cardiac puncture was carried out in each mouse to collect the blood samples. Afterward, the animals suffered a cervical dislocation. The necropsy was initiated with a midline incision into the abdomen; the colon and the feces were correctly removed.

### 2.4. Experimental Design

The mice were divided into six groups, according to the main objective of this experiment. The experimental groups were: the TNBS group (*n* = 10)—disease control group, where mice were only induced with colitis through the administration of TNBS, as previously described; TNBS + HEMIN 5 (*n* = 15) and TNBS + HEMIN 10 (*n* = 15) groups—treated groups, wherein animals were induced with colitis through the administration of TNBS over five weeks and treated daily with 5 mg/kg/day and 10 mg/kg/day of hemin by IP administration in the last two weeks, for fourteen days (between days 21–34 of the experiment), respectively; HEMIN 10 group (*n* = 10)—drug control group, wherein the mice were treated daily with only 10 mg/kg/day of hemin by IP administration for fourteen days; ethanol group (*n* = 10)—TNBS’s vehicle control group, wherein the mice received five IR administrations of 100 μL of the TNBS vehicle, which was ethanol 50%, one per week; and sham group (*n* = 10)—control group, wherein the mice received five IR administrations of 100 μL of NaCl 0.9%, one per week.

### 2.5. Monitoring of Clinical Signs

The mice were observed and monitored daily in terms of their body weight, morbidity, stool consistency, and anus appearance.

### 2.6. Biological Markers and Inflammatory Response

The biochemical markers analyzed were Alkaline Phosphatase (ALP), alanine aminotransferase (ALT), urea, and creatinine. To evaluate the inflammatory response, a pro-inflammatory cytokine, TNF-α, and an anti-inflammatory cytokine, interleukin (IL)-10, were measured and expressed as pg/mL. Centrifugation (3600 rpm for 15 min) was performed to separate the serum from the blood samples, and its analysis was possible using an automated clinical chemistry analyzer (ADVIA^®^1200). Fecal hemoglobin and fecal calprotectin were quantified in feces through a quantitative method by immunoturbidimetry (Kroma Systems, Gali, Índia) and ELISA, respectively.

### 2.7. Data Processing and Statistical Analysis

All of the results were expressed as mean ± SD of N observations, where N represents the number of animals analyzed. Data analysis was performed using GraphPad Prism 5.0 software (GraphPad, San Diego, CA, USA). The results were analyzed by a one-way ANOVA test to determine statistical significance between the control and experimental groups, followed by Tukey’s post hoc test for multiple comparisons. A *p*-value of less than 0.05 was considered significant.

## 3. Results

### 3.1. Clinical Signs of Illness

The mice from the TNBS group presented changes in intestinal motility, characterized by the presence of diarrhea and/or soft stool along with moderate edema of the anus. In the TNBS + HEMIN groups, considering both dosages, some of the mice also presented alterations in intestinal motility, but in a light form. The HEMIN 10, Sham, and Ethanol control groups remained without any alterations.

Regarding body weight changes (Figure 1), an increase in body weight was observed, in terms of percentage, in all the experimental groups, until the end of the experiment. The animals of the TNBS groups presented the lowest increase, gaining 4.95 ± 2.61% of their initial weight, while the animals of the Sham group presented a higher increase, gaining 15.15 ± 6.53% of their initial weight. The results of body weight between the TNBS + HEMIN treatment groups, at both dosages, were similar. At the end of the experimental period, the TNBS + HEMIN 5 groups gained 8.05 ± 3.79% of their initial weight and the TNBS + HEMIN 10 groups gained 10.23 ± 4.13% of their initial weight. The HEMIN 10 and ethanol groups gained 12.02 ± 5.33% and 10.71 ± 4.51% of their initial weights, respectively. There were no statistically significant differences between all groups, considering a *p* < 0.05. Therefore, it is not possible to determine a significant effect of hemin on the variation of the body weight between the treated and non-treated groups.

### 3.2. Biochemical Markers

Fecal calprotectin was measured and compared between all the experimental groups (Figure 2). This is a protein released by neutrophils when there is inflammation in the gastrointestinal tract [24]. The highest concentration of fecal calprotectin was noticed in the TNBS group (110.6 ± 15.14 ng/mg). The hemin treatment was able to decrease the fecal calprotectin concentration at both dosages, an effect that had statistical significance (*p* < 0.001 compared with the TNBS group). However, the differences between dosages of hemin were not statistically significant: the TNBS + HEMIN 5 group presented a dosage of 9.78 ± 2.39 ng/mg, whereas the TNBS + HEMIN 10 group displayed a dosage of 11.44 ± 3.28 ng/mg. The fecal calprotectin concentration in the HEMIN 10, sham, and ethanol control groups was very similar, at 7.00 ± 1.42 ng/mg, 6.50 ± 0.71 ng/mg and 9.00 ± 0.0 ng/mg, respectively (*p* < 0.001 compared with the TNBS group).

Fecal hemoglobin was measured and compared between all the experimental groups (Figure 3). This marker allowed for an evaluation of the intensity of hemorrhagic focus and the impact of hemin treatment. The mice in the TNBS group presented a more severe presence of blood in feces compared with the HEMIN 10 (6.93 ± 2.10 μmol/g feces vs. 1.17 ± 0.12 μmol/g feces, *p* < 0.001), sham (6.93 ± 2.10 μmol/g feces vs. 1.70 ± 0.20 μmol/g feces, *p* < 0.001), and ethanol (6.93 ± 2.10 μmol/g feces vs. 2.05 ± 0.50 μmol/g feces, *p* < 0.001) control groups. Hemin treatment demonstrated a statistically significant influence on the intensity of hemorrhagic focus as a significant difference in fecal hemoglobin was observed between the TNBS group, namely TNBS + HEMIN 5 (6.93 ± 2.10 μmol/g feces vs. 2.27 ± 0.54 μmol/g feces, *p* < 0.001), and TNBS + HEMIN 10 (6.93 ± 2.10 μmol/g feces vs. 2.43 ± 0.41 μmol/g feces, *p* < 0.001). However, the differences between dosages of hemin were not statistically significant.

ALP has a protective effect on the intestinal system, being responsible for the mucosal defense [16,21]. This marker was measured and compared between all the experimental groups (Figure 4). The highest concentration of ALP was noticed in the TNBS group (42.88 ± 5.69 U/L). The hemin treatment was able to decrease the ALP concentration at both dosages, an effect that had statistical significance (*p* < 0.001 compared with the TNBS group). However, the differences between dosages of hemin were not statistically significant: the TNBS + HEMIN 5 group presented a serum concentration of 31.56 ± 3.05 U/L, whereas the TNBS + HEMIN 10 group displayed a serum concentration of 32.67 ± 3.78 U/L. The ALP concentration in the ethanol group presented 30.50 ± 3.54 U/L (*p* < 0.01 compared with the TNBS group). The ALP concentration in the HEMIN 10 and sham groups was very similar, at 24.33 ± 1.53 U/L, and 24.67 ± 2.52 U/L, respectively (*p* < 0.001 compared with the TNBS group).

### 3.3. Measurement of Cytokines

In pathological conditions, pro-inflammatory cytokines, such as TNF-α, can become dysregulated, promoting an exacerbated inflammatory response. This cytokine was measured and compared between all the experimental groups (Figure 5). The TNBS group presented the highest concentration of this pro-inflammatory cytokine (76.96 ± 8.21 pg/mL). The treatment with hemin at both dosages, 5 mg/kg/day and 10 mg/kg/day, allowed the TNF-α levels to significantly decrease compared to the TNBS group (50.51 ± 5.04 pg/mL and 53.51 ± 5.66 pg/mL, respectively; *p* < 0.001). However, the differences between dosages of hemin were not statistically significant. The HEMIN 10, sham, and ethanol control groups also demonstrated a significant decrease in this cytokine compared to the TNBS group (31.80 ± 3.38 pg/mL, 41.53 ± 2.15 pg/mL, and 41.50 ± 3.96 pg/mL, respectively; *p* < 0.001).

IL-10 was measured and compared between all of the experimental groups (Figure 6). This is an anti-inflammatory cytokine and in pathological conditions, this cytokine can also become dysregulated. The TNBS group presented the highest concentration of this pro-inflammatory cytokine (70.31 ± 6.88 pg/mL). A significant decrease in the concentration of IL-10 was noticed in the treatment groups at both dosages, compared to the TNBS group (*p* < 0.001). However, the differences between dosages of hemin were not statistically significant: the TNBS + HEMIN 5 group presented a dosage of 38.27 ± 4.93 pg/mL, whereas the TNBS + HEMIN 10 group displayed a dosage of 44.24 ± 8.79 pg/mL. The HEMIN 10, sham, and ethanol control groups also demonstrated a significant decrease in this cytokine compared to the TNBS group (27.97 ± 4.08 pg/mL, 33.37 ± 1.82 pg/mL, and 38.60 ± 7.21 pg/mL, respectively; *p* < 0.001).

### 3.4. Hepatic and Renal Functions

The hepatic function was evaluated based on the ALT concentration in the plasma of the mice in all of the experimental groups (Figure 7). The TNBS group showed the highest concentration of ALT (30.88 ± 4.64 U/L). The treatment with hemin at both dosages, 5 mg/kg/day and 10 mg/kg/day, promotes a decrease in ALT levels with statistical significance compared to the TNBS group (20.33 ± 3.67 U/L; *p* < 0.001 and 22.56 ± 3.61 U/L; *p* < 0.01, respectively). However, the differences between dosages of hemin were not statistically significant. The HEMIN 10 and sham control groups demonstrated a significant decrease in this marker compared to the TNBS group (22.67 ± 4.04 U/L; *p* < 0.05 and 20.00 ± 1.00 U/L; *p* < 0.01, respectively). The ethanol group did not demonstrate a significant decrease in ALT levels (22.00 ± 4.24 U/L).

In order to evaluate renal function, the concentrations of renal damage markers, such as urea and creatinine, were measured and compared between all the experimental groups. (Figure 8 and Figure 9). Firstly, in terms of the concentration of urea, there were no statistically significant differences between experimental groups. However, the TNBS + HEMIN 5 and TNBS + HEMIN 10 groups presented the highest value, with 81.78 ± 2.86 mg/dL, and 83.67 ± 4.42 mg/dL, respectively. The HEMIN 10 group showed the lowest value (69.00 ± 2.65 mg/dL, *p* < 0.05 compared with the TNBS group). The TNBS, sham, and ethanol groups displayed similar results between them, at 79.13 ± 7.49 mg/dL, 80.33 ± 1.53 mg/dL, and 79.00 ± 5.66 mg/dL, respectively.

Second, in terms of the concentration of creatinine, the TNBS group showed the highest concentration of this marker (0.55 ± 0.04 mg/dL). A significant decrease in the concentration of creatinine was noticed in the treatment groups at both dosages, compared to the TNBS group (*p* < 0.001). However, the differences between dosages of hemin were not statistically significant: the TNBS + HEMIN 5 group presented a dosage of 0.36 ± 0.04 mg/dL, whereas the TNBS + HEMIN 10 group displayed a dosage of 0.40 ± 0.02 mg/dL. The HEMIN 10 and sham control groups demonstrated a significant decrease in the concentration of creatinine compared to the TNBS group (0.33 ± 0.01 mg/dL and 0.38 ± 0.09 mg/dL; *p* < 0.001, respectively). The ethanol group did not demonstrate a significant decrease in creatinine levels (0.45 ± 0.04 mg/dL).

## 4. Discussion

Hemin is well known to induce HO-1. HO-1 is a rate limiting enzyme of heme metabolism and produces antioxidant and anti-inflammatory products such as biliverdin, free iron, and carbon monoxide [16,25,26]. Thus, a better understanding of the HO system may result in new therapeutic strategies for various pathologies. In fact, HO-1 can confer cytoprotective, anti-apoptotic, and anti-inflammatory properties, thus becoming a possible pharmacological approach for the management of various gastrointestinal diseases, namely IBD [16,25].

Concretely, the data available regarding the utilization of hemin in IBD have shown a positive influence, which is demonstrated by the reduction in the expression of fecal hemoglobin, ALP, myeloperoxidase, and pro-inflammatory cytokines [16]. The preclinical study that evaluated the effect of hemin in IBD developed an acute model of colitis, and the experiment took 4 days in total. In light of this, the objective of this experiment was to evaluate the same molecule at the same dosages, but now in a chronic animal model of IBD.

During the experimental study, the mice with TNBS-induced colitis revealed an alteration of intestinal motility that suggests a correct induction of experimental colitis [27]. However, the treatment with hemin was able to attenuate diarrhea and moderate edema of the anus compared to the non-treated mice. Regarding changes in body weight, although we noticed a percentage gain, the groups treated with hemin did not present significant differences from the TNBS group. In this sense, the hemin did not demonstrate a significant effect on the body weight of the animals.

Fecal calprotectin is released by neutrophils when there is inflammation in the gastrointestinal tract [24,28,29,30]. The mice with TNBS-induced colitis showed higher concentrations of fecal calprotectin compared to the healthy mice, and the fecal calprotectin levels were significantly reduced by hemin treatment in both dosages. However, there was no dose-dependent effect between the two dosages tested.

Stools containing blood are a characteristic of IBD [31]. The determination of fecal hemoglobin allows for an evaluation of the intensity of the hemorrhagic focus, can be useful in the detection of lesions accompanied by bleeding, and can verify whether the hemin treatment has an influence on the amelioration of the bloody stools [28]. It is possible to confirm that the treatment with hemin had a positive influence on decreasing the levels of fecal hemoglobin in both dosages, compared to the TNBS group. However, there was no dose-dependent effect between the two dosages tested. These results, to some extent, agree with previous studies that also used hemin, although there is not the same similarity with regard to the question of the dose-dependent effect [16].

The intestinal ALP is expressed on the enterocytes and is responsible for mucosal defense, so in the case of colon lesions, an increase in this marker is expected [28]. The mice with TNBS-induced colitis showed higher concentrations of ALP compared to the healthy mice, and the ALP levels were significantly reduced by hemin treatment, suggesting that hemin has an important anti-inflammatory effect, which has been evidenced in previous studies [16]. However, there was no dose-dependent effect between the two dosages tested.

In order to determine whether the treatment with hemin may modulate the inflammatory process through the regulation of the secretion of pro-inflammatory and anti-inflammatory cytokines, the levels of TNF-α and IL-10, respectively, were analyzed. Regarding TNF-α, the mice with TNBS-induced colitis exhibited a significant increase in this pro-inflammatory cytokine. However, the treatment with hemin in both dosages significantly decreased the level of this cytokine, confirming the anti-inflammatory effect of Hemin that has been described in the literature [16]. Among the two dosages tested, there were no statistically significant differences, so there was no dose-dependent effect. Concerning IL-10, it is possible to confirm that the treatment with hemin had a positive influence on decreasing the levels of this anti-inflammatory cytokine in both dosages in comparison to the TNBS group, demonstrating that hemin is beneficial in the inflammatory response caused by colitis. However, there was no dose-dependent effect between the two dosages tested.

As IBD can promote extraintestinal manifestations, the periodic evaluation of hepatic and renal functions should be emphasized. So, the determination of plasma ALT, urea, and creatinine levels helps us to understand this, and allows for the evaluation of the safety profile of the applied treatment. Indeed, the treatment with hemin allowed for better results to be obtained in these parameters, except for the urea levels, in which the differences were not significant in relation to the TNBS group. These results agree with previous studies that also used hemin [16]. It is also possible to conclude that hemin does not promote renal and/or hepatic changes as adverse drug reactions, because the HEMIN10 group had no elevated levels of these biochemical markers.

According to the results obtained in this study, it is possible to observe the beneficial role of hemin treatment in a mouse model of colitis. The treatment with hemin reduces intestinal damage, hemorrhagic focus, and tissue inflammation, without significant side effects.

## 5. Conclusions

It is possible to observe the anti-inflammatory effect of hemin in a TNBS-induced animal model of colitis. This pharmaceutical approach promoted a reduction in ALP, fecal hemoglobin, fecal calprotectin, TNF-α, IL-10, ALT, and creatinine. Indeed, the treatment with hemin decreases the severity of the disease because of its ability to improve several inflammation markers, suggesting hemin has an anti-inflammatory effect by HO-1 induction. Additionally, hemin demonstrated a beneficial effect on the extraintestinal manifestations, such as the hepatic function, and did not show any significant adverse effects upon its use. These findings suggest that hemin appears to significantly inhibit the chronic inflammatory response in this experimental colitis. In sum, according to the results obtained in this experiment, it is possible to conclude that the treatment with hemin is a very promising approach for the future management of chronic inflammatory diseases, such as IBD.

## Figures and Tables

**Figure 1 biomedicines-10-02025-f001:**
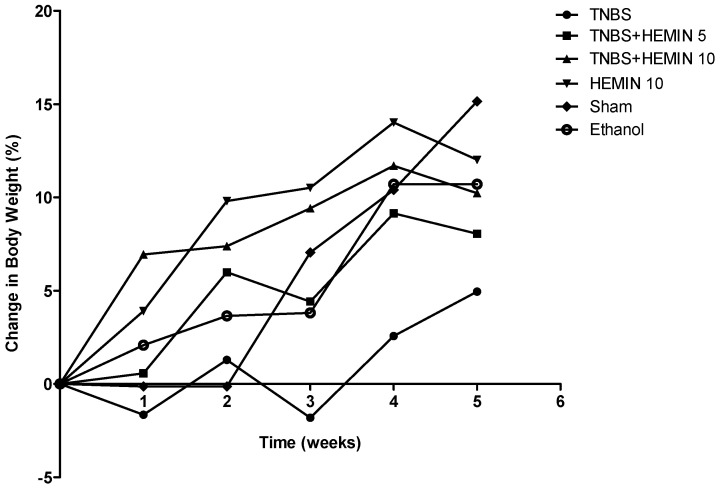
Change in body weight during hemin treatment in the IBD.

**Figure 2 biomedicines-10-02025-f002:**
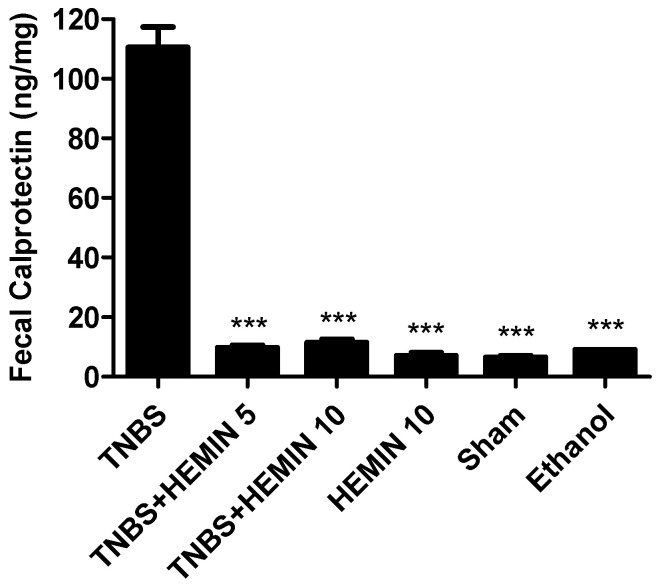
Effect of hemin treatment on fecal calprotectin in the IBD. Legend: One-way ANOVA for multiple comparisons and Tukey’s post hoc test. *** *p* < 0.001; compared with TNBS group.

**Figure 3 biomedicines-10-02025-f003:**
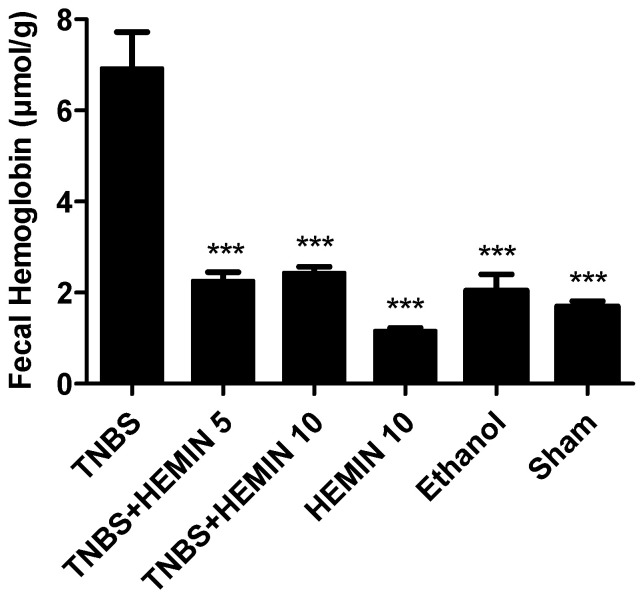
Effect of hemin treatment on fecal hemoglobin in the IBD. Legend: One-way ANOVA for multiple comparisons and Tukey’s post hoc test. *** *p* < 0.001; compared with TNBS group.

**Figure 4 biomedicines-10-02025-f004:**
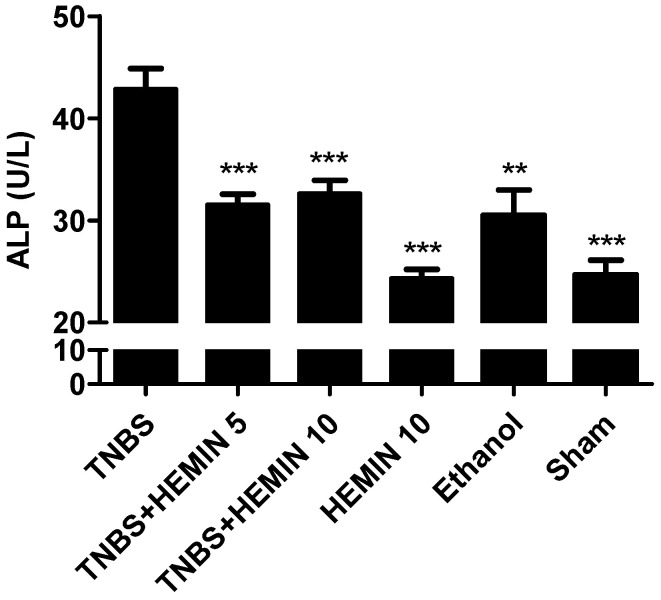
Effect of hemin treatment on ALP concentration in the IBD. Legend: One-way ANOVA for multiple comparisons and Tukey’s post hoc test. ** *p* < 0.01; *** *p* < 0.001; compared with TNBS group.

**Figure 5 biomedicines-10-02025-f005:**
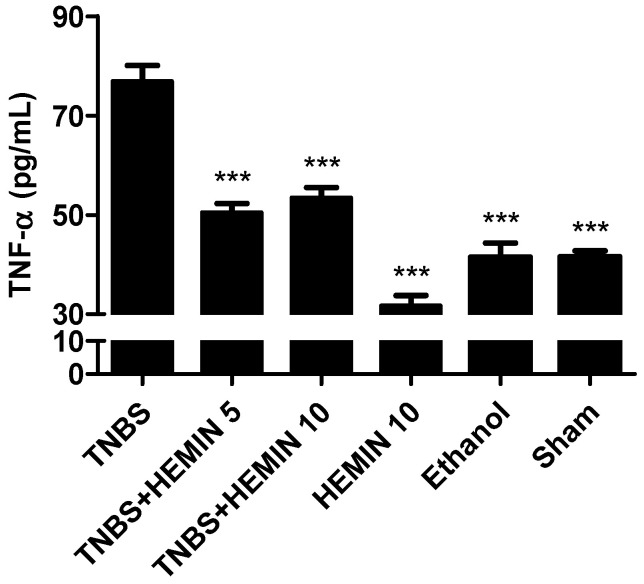
Effect of hemin treatment on TNF-α concentration in the IBD. Legend: One-way ANOVA for multiple comparisons and Tukey’s post hoc test. *** *p* < 0.001; compared with TNBS group.

**Figure 6 biomedicines-10-02025-f006:**
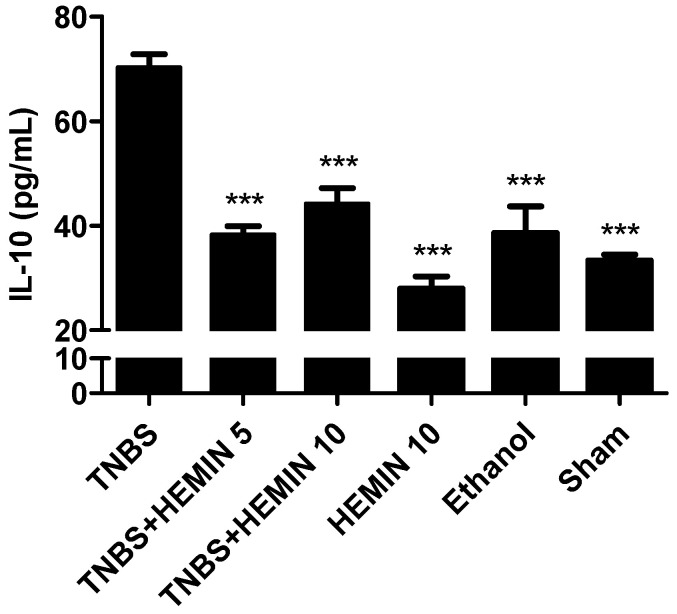
Effect of hemin treatment on IL-10 concentration in the IBD. Legend: One-way ANOVA for multiple comparisons and Tukey’s post hoc test. *** *p* < 0.001; compared with TNBS group.

**Figure 7 biomedicines-10-02025-f007:**
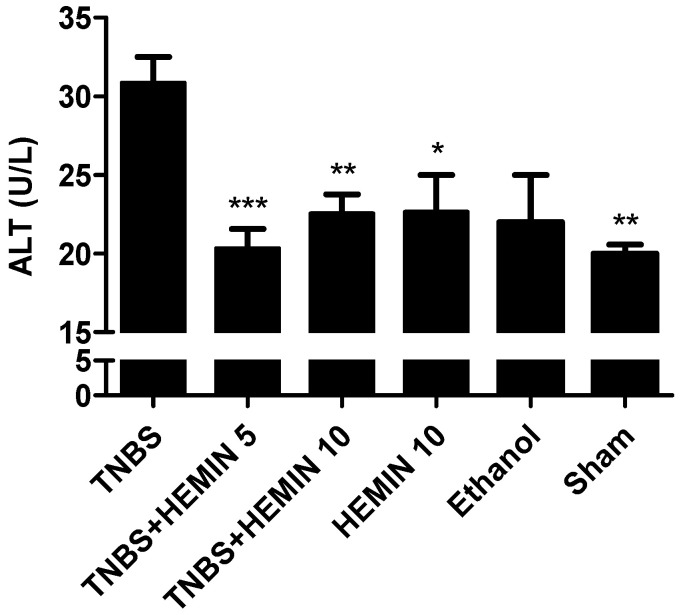
Effect of hemin treatment on ALT concentration in the IBD. Legend: One-way ANOVA for multiple comparisons and Tukey’s post hoc test. * *p* < 0.05; ** *p* < 0.01; *** *p* < 0.001; compared with TNBS group.

**Figure 8 biomedicines-10-02025-f008:**
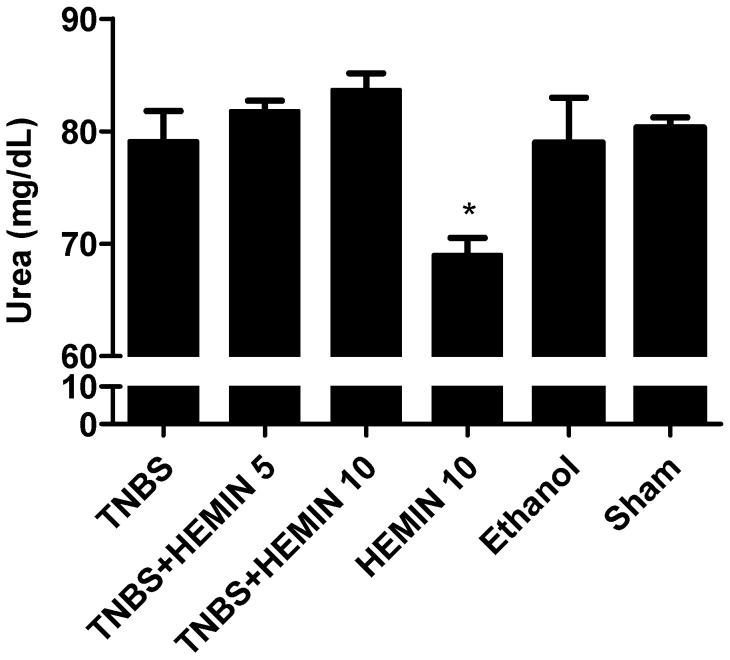
Effect of hemin treatment on urea concentration in the IBD. Legend: One-way ANOVA for multiple comparisons and Tukey’s post hoc test. * *p* < 0.05; compared with TNBS group.

**Figure 9 biomedicines-10-02025-f009:**
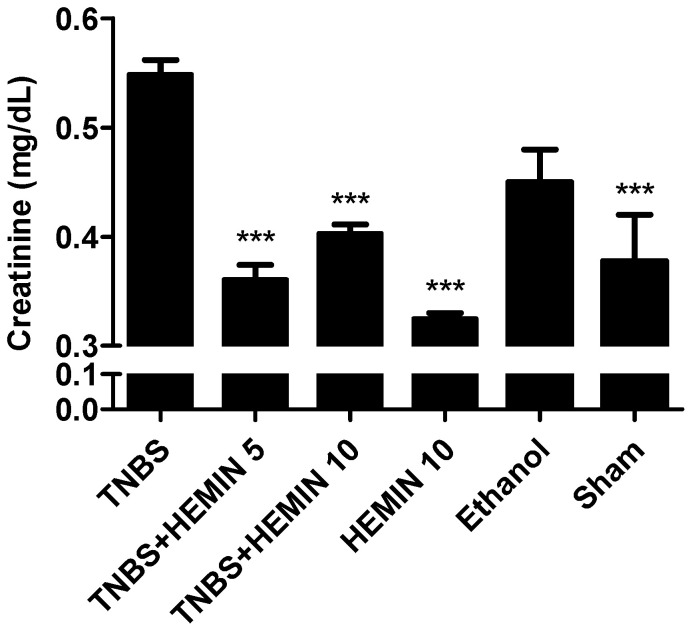
Effect of hemin treatment on creatinine concentration in the IBD. Legend: One-way ANOVA for multiple comparisons and Tukey’s post hoc test. *** *p* < 0.001; compared with TNBS group.

## Data Availability

Not applicable.

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
