# Peer review of "Hemin Ameliorates the Inflammatory Activity in the Inflammatory Bowel Disease: A Non-Clinical Study in Rodents"

_biomedicines, 2022, doi:10.3390/biomedicines10082025_

Round 1
Reviewer 1 Report
In this manuscript the authors tested the efficacy of hemin in treating TNBS-induced colitis in mice. This is building on the group’s previous finding in 4-days treatment model; here, in a 5-weeks TNBS-induced colitis model, the authors showed that hemin at two doses (5 and 10 mg/kg/day) significantly reduced fecal calprotectin and hemoglobin, suggesting amelioration of colitis by hemin. While the data are interesting, there are several concerns regarding data presentation.
Major comments:
- The description of the experimental design is not entirely clear. One assumes the TNBS-induction was for 5 weeks and hemin was given in the last 2 weeks of the 5-weeks study period? When are feces collected to test for calprotectin and hemoglobin levels?
- What is the n number for each group?
- On that note, there are no error bars in Fig. 1. Also, for readers to better gage the spread of data, it may be better to present data in individual points for each bar.
- For TNFa and IL-10 levels, the authors mentioned colon levels (line 312), but the units are pg/ml, not per mg tissue protein? Please clarify.
- Similarly, for ALP levels, please clarify whether tissue or serum levels were assessed.
- In discussion, the authors mentioned that intestinal motility were ameliorated by hemin treatment (lines 284-286); however, no data were shown to support this. It was only described briefly in the first paragraph in Results. Have the authors scored fecal consistency and bleeding to measure disease activities?
- For Introduction, can the authors explain a little more on the role and regulation of HO-1 expression in colitis? Is HO-1 expression elevated during disease progression? Is this part of defense response of the organism to resolve inflammation?
Minor comments:
- Lines 56 and 58: these statements seemed repetitive.
- What might be the target cells for hemin? E.g. intestinal epithelial cells? Neutrophils, macrophages, or other immune cell types?
Author Response
Dear Dr,
Thank you for your feedback and all the time you spent working on our manuscript.
The English were revised, and we already improved the manuscript as you can verify, namely the Introduction with the role of HO-1 and the target cells for hemin.
- For readers to better gage the spread of data, it may be better to present data in individual points for each bar.
Response:
Our research group works in the acute model (Mateus et al, 2016; Mateus et al, 2018; Mateus et al, 2018)) and in the chronic model of TNBS-induced colitis in mice (Silva et al, 2022). Graphically, and in order to facilitate the interpretation of the results, we decided to make a daily graph for the acute model (developed in 4 days) and a weekly graph for the chronic model (developed in 5 weeks). We considered that this was the graph that best represented the weight variation of the mice over the weeks. Concretely, in terms of interpretation, the absence of weekly statistically significant differences between TNBS and Sham groups in the chronic model can be explained by the lower concentration of TNBS used (1% in the chronic model and 2.5% in the acute model) and because mice gain resistance to TNBS along the time. In the first three weeks of the experiment, the TNBS group showed decreases in body weight. This decrease may be related to the fragility of the mice following the first three TNBS administrations. However, a recovery in body weight occurred over the following weeks corresponding to an adaptation.
- For TNFa and IL-10 levels, the authors mentioned colon levels (line 312), but the units are pg/ml, not per mg tissue protein? Please clarify.
Response:
It was a lapse in line 312. Thanks for the detection. The pro and anti-inflammatory cytokines were analyzed in the serum.
– Similarly, for ALP levels, please clarify whether tissue or serum levels were assessed.
Response:
The ALP was also assessed in the serum.
- In discussion, the authors mentioned that intestinal motility were ameliorated by hemin treatment (lines 284-286); however, no data were shown to support this. It was only described briefly in the first paragraph in Results. Have the authors scored fecal consistency and bleeding to measure disease activities?
Response:
For five weeks, we monitor daily the fecal consistency but we don’t score nor measure the disease activity. However, we consider that the evaluation of the disease activity index would be beneficial to take into account in future experiments, since it is a score that combines three clinical signs, such as weight loss, stool consistency, and fecal blood, which is commonly used in non-clinical studies about IBD.
Sincerely yours,
Prof. Vanessa Alexandra Pinho Mateus, BPharm MSc PhD
Professor of Pharmacology and Pharmacotherapy
Diretor of Pharmacy Degree Course in Lisbon School of Health Technology (Polytechnic Institute of Lisbon)
Member of Coordinating Committee of Health and Technology Research Center (H&TRC)

Reviewer 2 Report
The manuscript assessed the possibility of Hemin as a treatment for IBD, which were induced by TNBS application, using mice. Mice were divided into 6 groups, including TNBS, TNBS + HEMIN 5, TNBS + HEMIN 10, HEMIN 10, Sham and Ethanol groups. Clinical illness was observed, body weights were tracked, and fecal calprotectin, fecal hemoglobin, ALP, TNF-α, IL-10, ALT, urea, creatinine were measured. The results indicate TNBS group presented changes in intestinal motility, TNBS + HEMIN groups presented lighter alterations in intestinal motility, and the HEMIN 10, Sham, and Ethanol control groups remained without any alterations. Although there is no change in body weight among groups, calprotectin, hemoglobin, ALP, inflammation parameters, indicator of hepatic function ALT and indicator of renal function creatinine all showed changes in TNBS group, which can be reversed by HEMIN application, as shown by TNBS + HEMIN 5 and TNBS + HEMIN 10 groups. The authors concluded that hemin reduced the chronic inflammatory response, with extraintestinal benefits. Treatment with hemin can be a very promising approach for the future management of chronic inflammatory diseases.
Overall, the manuscript is clear, well-structured and scientifically sound. The statistic and data are presented in detail. The logic underlying each measurement is easy to understand. The followings are three concerns about the paper:
1. The authors mentioned in Introduction and Discussion that Hemin can induce HO-1. The role of HO-1 was emphasized and it seems that the authors believe the effect of Hemin they found here is through HO-1. However, there is no measurement of HO-1 in the manuscript.
2. They authors did not find significant changes in weight gain. The difference between TNBS and Sham group seems to be pretty big though. Did they only compare the last weeks data? It is recommended the authors consider other statistical methods, like generalized linear regression, to include all time points and analyze the data again.
3. Figure 2-4 all assessed the intestinal health, and can be combined; Figure 5-6 assessed inflammation, and can be combined; Figure 8-9 are about renal function, and can be combined. Plus, bar-plot is not a good way for data presentation. It is recommended to show individual data points in the figure.
Author Response
Dear Dr,
Thank you for your feedback and all the time you spent working on our manuscript.
The English were revised.
- The authors mentioned in Introduction and Discussion that Hemin can induce HO-1. The role of HO-1 was emphasized and it seems that the authors believe the effect of Hemin they found here is through HO-1. However, there is no measurement of HO-1 in the manuscript.
Response:
Currently, hemin is used in the treatment of acute attacks of hepatic porphyria. Porphyrias are characterized by the existence of an enzymatic blockage in the heme biosynthesis pathway resulting in a lack of heme necessary for the synthesis of various hemoproteins or in the accumulation before the metabolic blockade of heme precursors, which are directly or indirectly toxic to the organism. The administration of hemin reduces heme deficiency by suppressing the activity of delta-aminolevulinic acid synthases (the main enzyme in the synthesis of porphyrins) by retro-regulation, which reduces the production of porphyrins and toxic precursors of heme. In this way, helping to restore normal levels of hemoproteins and respiratory pigments, heme corrects the biological disturbances seen in patients with porphyria (Bonkowsky et al, 1971).
HO-1 is an enzyme that is involved in the catabolism of the heme group, having a cytoprotective action capable of inhibiting the inflammatory response and reducing oxidative stress. HO-1 expression can be induced by Hemin. Proteins that contain a heme group (hemoglobin, myoglobin, cytochrome p450, inducible nitric oxide synthase (iNOS), among others) play a fundamental role in the physiological processes related to oxygen transport, mitochondrial respiration, and signal transduction. Because the free heme group exerts cytotoxic action by forming free oxygen radicals and promoting lipid peroxidation, this group is degraded by the enzyme heme-oxygenase. This enzyme is expressed in the constitutive form, heme-oxygenase 2 (HO-2), which regulates the normal functioning of cells, and in the induced form, HO-1, which is expressed in the form of a response to tissue damage. Degradation of the heme group by HO-1 will result in equimolar amounts of carbon monoxide (CO), Fe2+ and biliverdin. Later, the biliverdin is reduced to bilirubin by biliverdin reductase (Kirkby et al, 2006).
It has been shown that HO-1 and HO-1 mRNA expression is remarkably elevated in situations of intestinal inflammation when compared to healthy intestines. Furthermore, in situations of mucosal inflammation in active ulcerative colitis, HO-1 protein expression is also elevated. This evidence suggests that the increase in HO-1 expression is due to inflammation. HO-1 expression occurs essentially in macrophages, as in previous histological studies the result was CD-68 positive. Although the expression is mostly observed in macrophages, other studies have described the localization of HO-1 in the colonic mucosa, which, in addition to being expressed in inflammatory cells, is also expressed in epithelial cells (Naito et al, 2011).
Recent studies demonstrate that the degradation of the heme group by HO-1 has a cytoprotective role in tissues, namely its by-products that have an anti-inflammatory, anti-apoptotic, anti-oxidant and immunomodulatory action (Naito et al, 2011; Otterbein et al, 2003).
When exposure to CO long-term and in high concentrations, it has a toxic effect. On the other hand, when exposure occurs for a short period of time and at low concentrations of CO, it has a vasodilating, anti-apoptotic and anti-inflammatory effect. These effects are verified in the degradation of the heme group. The anti-inflammatory action of CO comes from its ability to mediate the mitogen-activated protein kinase (MAPK) process, namely that associated with the p38 gene, which is a physiological process that is associated with responses to stress signals. CO, by potentiating this process, results in an anti-inflammatory action, by decreasing TNF-α. So in our work we measured this pro-inflammatory cytokine to evaluate the anti-inflammatory effect.
- They authors did not find significant changes in weight gain. The difference between TNBS and Sham group seems to be pretty big though. Did they only compare the last weeks data? It is recommended the authors consider other statistical methods, like generalized linear regression, to include all time points and analyze the data again.
Response:
Our research group works in the acute model (Mateus et al, 2016; Mateus et al, 2018; Mateus et al, 2018)) and in the chronic model of TNBS-induced colitis in mice (Silva et al, 2022). During the daily monitorization of the animals, throughout the entire experience, we found statistically significant differences between groups Sham and TNBS in both models. Graphically, and in order to facilitate the interpretation of the results, we decided to make a daily graph for the acute model (developed in 4 days) and a weekly graph for the chronic model (developed in 4 weeks). In this sense, the absence of weekly statistically significant differences between these two groups in the chronic model can be explained by the lower concentration of TNBS used (1% in the chronic model and 2.5% in the acute model) and because mice gain resistance to TNBS along the time. In the first three weeks of the experiment, the TNBS groups showed decreases in body weight. This decrease may be related to the fragility of the mice following the first three TNBS administrations. However, a recovery in body weight occurred over the following weeks corresponding to an adaptation.
- Figure 2-4 all assessed the intestinal health, and can be combined; Figure 5-6 assessed inflammation, and can be combined; Figure 8-9 are about renal function, and can be combined. Plus, bar-plot is not a good way for data presentation. It is recommended to show individual data points in the figure.
Response:
We agreed with the reviewer and in this sense, we created subpoints that precisely matched the graphics suggested by the reviewer. However, if we place them side by side, the graphs will be small and difficult to read and interpret. In the author’s opinion, these bar graphs are the ones that best translate the results.
Sincerely yours,
Prof. Vanessa Alexandra Pinho Mateus, BPharm MSc PhD
Professor of Pharmacology and Pharmacotherapy
Diretor of Pharmacy Degree Course in Lisbon School of Health Technology (Polytechnic Institute of Lisbon)
Member of Coordinating Committee of Health and Technology Research Center (H&TRC)
